# Industrial Benchmarking of LLMs: Assessing Hallucination in Traffic Incident Scenarios with a Novel Spatio-Temporal Dataset

## Abstract

Large language models (LLMs) hold revolutionary potential to digitize and enhance the Health & Public Services (H&PS) industry. Despite their advanced linguistic abilities, concerns about accuracy, stability, and traceability still persist, especially in high-stakes areas such as transportation systems. Moreover, the predominance of English in LLM development raises questions about how they perform in non-English contexts. This study introduces a novel cross-lingual benchmark dataset comprising nearly 99,869 real traffic incident records from Vienna (2013-2023) to assess the robustness of state-of-the-art LLMs ($\geq 9$) in the spatio and temporal domain of traffic incident classification. We then explored three hypotheses — sentence indexing, date-to-text conversion, and German-to-English translation — and incorporated Retrieval Augmented Generation (RAG) to further examine the models' ability to handle hallucinations in both spatial and temporal contexts. Our experiments with GPT-4 and Llama models reveal significant performance disparities across these hypotheses in the spatio-temporal domain and also demonstrate how RAG can mitigate what types of hallucinations. These findings underscore the need for enhanced cross-lingual capabilities and improved explainability in LLMs. We provide open access to our Health & Public Services (H&PS) traffic incident dataset, with the project demo and code available at Website `https://sites.google.com/view/llmhallucination/home`.

## 1 Introduction

Large Language Models (LLMs) such as GPT-3.5 (Radford et al., 2018), GPT-4 (Achiam et al., 2023), and LaMDA (Thoppilan et al., 2022) have substantially enhanced public access to complex information, particularly in sectors such as healthcare and public services (H&PS). These models are celebrated for their capability to demystify intricate information, assisting in tasks ranging from routine inquiries to aiding clinical decision-making (Brown et al., 2020; Radford et al., 2018; Achiam et al., 2023; Thoppilan et al., 2022). ChatGPT (Achiam et al., 2023), a derivative of the InstructGPT model (Ouyang et al., 2022), has garnered a vast user base, especially for textual tasks, through its sophisticated multi-turn prompting dialog interface, which is further refined by Reinforcement Learning with Human Feedback (RLHF) (Lambert et al., 2022). Despite its proficiency in various NLP tasks, ChatGPT has faced also critique. Anecdotal reports on ChatGPT revealed consistently remaining challenges (Bang et al., 2023) - for instance, it struggles with specific reasoning tasks (Davis, 2023; Guo et al., 2023), often hallucinates facts, and produces non-factual statements, undermining its reliability (Shen et al., 2023; Thorp, 2023). Additionally, its language coverage remains limited and its predominant focuses on English in model training and evaluation raises issues of equitable access for non-English speakers (Seghier, 2023), especially given that over 82% of the global population does not speak English as their primary or secondary language (Crystal, 2003; Lu et al., 2022; Jiao et al., 2023).

Previous research has primarily focused on evaluating the knowledge capabilities of LLMs (Singhal et al., 2022) or leveraging the stochasticity in sampled responses to assess the factuality of generated outputs (Goodrich et al., 2019). Furthermore, substantial efforts have been directed towards developing neural network techniques to make accurate predictions on synthetic data (Li et al., 2024). Given that LLMs are trained on extensive datasets, it is crucial to explore how these models perform

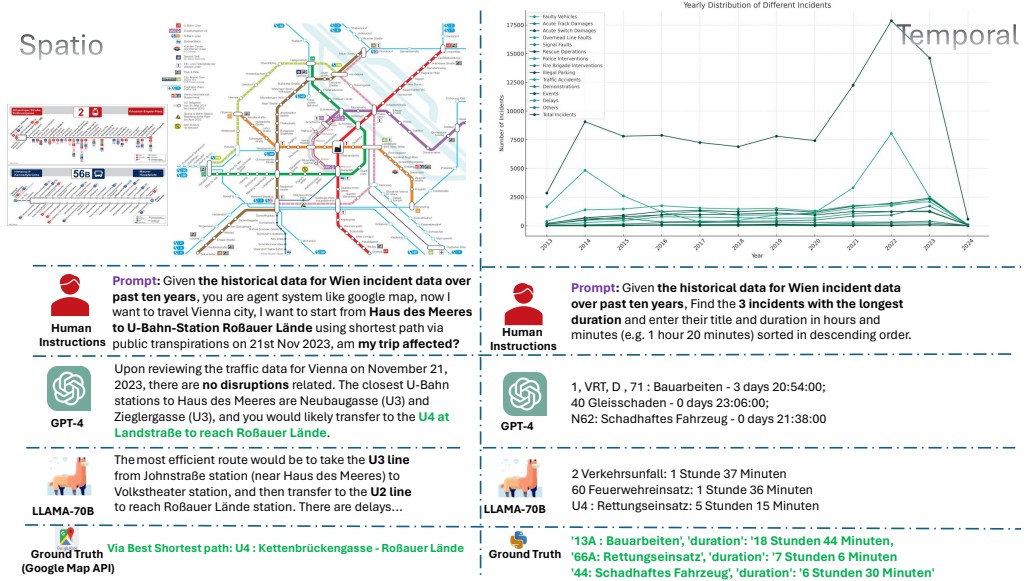

Figure 1: The H&PS Traffic Incidents Dataset includes 99,869 recorded incidents within the Vienna public transportation system, categorized into 14 distinct scenarios: Faulty Vehicles, Acute Track Damages, Acute Switch Damages, Overhead Line Faults, Signal Faults, Rescue Operations, Police Interventions, Fire Brigade Interventions, Illegal Parking, Traffic Accidents, Demonstrations, Events, Delays, and Other Incidents.

with real industrial spatio-temporal data and to understand variations in performance across different spatio-temporal contexts.

To address these challenges, our study introduces a novel, comprehensive multilingual benchmark from the industry for evaluating LLMs in sensitive sectors such as health and public services across spatio-temporal domains. We aim to bridge the language equity gap and evaluate the stability of these models. Our contributions include:

- The creation of an H&PS Traffic Incidents Spatio-Temporal Dataset, containing diverse traffic incidents over a decade, totaling nearly 99,869 records, for investigating LLM hallucinations (Brown et al., 2020).
- A robust quantitative analysis of three hypotheses across multiple languages aimed at enhancing the performance of state-of-the-art (SOTA) LLMs. This analysis underscores their practical utility and identifies their vulnerabilities in managing real-world generative AI applications.
- An in-depth examination using Retrieval-Augmented Generation (RAG) (Jiang et al., 2023) to assess the influence of spatio-temporal data and prompts, providing unique insights derived from our extensive multilingual dataset.

## 2 RELATED WORK

In this section, we discuss previous works that investigate LLMs' (Brown et al., 2020) performance on hallucination resolution (Rawte et al., 2023). Previous studies have explored the capabilities of models like ChatGPT (Achiam et al., 2023), suggesting various methods to mitigate its limitations. For instance, Bang et al. (Bang et al., 2023) show that ChatGPT excels in zero-shot learning across 9 of 13 NLP datasets (Socher et al., 2013), even outperforming fully fine-tuned, task-specific language models in four different tasks. However, they also report a noticeable performance decline when handling languages with limited resources, particularly in non-Latin scripts. The weakness lies more in generation than in the understanding part of the translation process. Manakul et al. (Manakul et al., 2023) introduced SELFCHECKGPT, a method for detecting hallucinations in LLM-generated

responses. This technique, relying on response consistency, may overlook cases where LLMs deliver consistent but inaccurate information, leading to potential false negatives in hallucination detection.

Among related work, the XLingEval framework by Choudhury et al. (Choudhury et al., 2023) most closely aligns with our research. It assesses LLM behavior across several languages (English, Hindi, Chinese, and Spanish), focusing on metrics like correctness, consistency, and verifiability. Their findings indicate significant performance disparities across languages, with non-English responses generally showing an 18.12% decrease in quality (Choudhury et al., 2023). However, the study also shows limitations: it only investigated the influence of multilingualism with state-of-the-art LLMs, without further exploring how to avoid hallucinations in industries other than healthcare, or in languages such as German. Additionally, it did not examine the impact of input data on these LLMs or other factors beyond language type differences. For example, the format of the data, the effects of prompts under different temperature hyperparameters, and the interactions between prompts and data were not considered. In contrast, the UrbanGPT study (Li et al., 2024) utilizes LLMs specifically for modeling urban environments, applying a GPT variant to zero-shot learning tasks in traffic management and public safety. The study emphasizes the critical role of high-quality, representative spatio-temporal data in training effective models (Li et al., 2024).

We also acknowledge the strengths and weaknesses of RAG (Retrieval-Augmented Generation), as highlighted in recent studies. For instance, RAG may struggle to properly contextualize or synthesize retrieved data, leading to augmentations that either lack depth or fail to capture the nuances of the query (Jiang et al., 2023; Siriwardhana et al., 2023), particularly with non-English data (Jiang et al., 2023). Moreover, RAG systems can sometimes generate responses that are misleading, incomplete, or contextually off-target. Another concern is latency sensitivity, and training local LLMs with RAG is technically more complex and costlier than methods like prompt fine-tuning or data augmentation (Karpukhin et al., 2020; Guu et al., 2020). In this paper, we further explore both the strengths and limitations of RAG in handling temporal and spatial data within real-world industry applications.

Our work extends these discussions by adopting a quantitative methodology to rigorously test each hypothesis and by constructing a hybrid dataset of public traffic incident records from Vienna, spanning over decades. Distinguished from the XLingEval framework (Choudhury et al., 2023),it offers more realistic basis for investigating LLM hallucinations and enables more intricate spatio-temporal analyses, focusing particularly on minority languages like German.

## 3 H&PS TRAFFIC INCIDENTS DATASET

Table 1: Incident Statistics Per Year (2013*-2023). *Collection remained for 2013, 14th Sep - Dec.

| Incident Type | 2013* | 2014 | 2015 | 2016 | 2017 | 2018 | 2019 | 2020 | 2021 | 2022 | 2023 |
|---|---|---|---|---|---|---|---|---|---|---|---|
| Faulty Vehicles | 132 | 477 | 592 | 966 | 1282 | 921 | 949 | 1062 | 1527 | 1753 | 2326 |
| Acute Track Damages | 11 | 46 | 38 | 63 | 48 | 53 | 32 | 54 | 50 | 54 | 70 |
| Acute Switch Damages | 4 | 11 | 17 | 58 | 59 | 68 | 41 | 45 | 57 | 69 | 100 |
| Overhead Line Faults | 16 | 69 | 77 | 94 | 104 | 111 | 102 | 108 | 58 | 100 | 100 |
| Signal Faults | 2 | 20 | 21 | 45 | 25 | 27 | 28 | 41 | 20 | 48 | 65 |
| Rescue Operations | 198 | 701 | 912 | 1247 | 1341 | 1224 | 1378 | 1188 | 1693 | 1955 | 2413 |
| Police Interventions | 54 | 266 | 442 | 783 | 759 | 702 | 653 | 679 | 1062 | 1326 | 1289 |
| Fire Brigade Interventions | 17 | 84 | 152 | 267 | 274 | 305 | 325 | 287 | 325 | 332 | 403 |
| Illegal Parking | 137 | 507 | 775 | 953 | 975 | 1017 | 1047 | 1139 | 1236 | 1362 | 1229 |
| Traffic Accidents | 394 | 1386 | 1466 | 1749 | 1549 | 1457 | 1528 | 1292 | 1761 | 1879 | 2102 |
| Demonstrations | 0 | 25 | 40 | 44 | 40 | 89 | 127 | 142 | 215 | 239 | 252 |
| Events | 0 | 0 | 0 | 0 | 70 | 71 | 107 | 141 | 142 | 107 | 81 |
| Delays | 1675 | 4838 | 2608 | 1213 | 468 | 944 | 1137 | 3320 | 8048 | 8408 | 2502 |
| Other Incidents | 220 | 651 | 655 | 339 | 490 | 394 | 503 | 647 | 943 | 647 | 1724 |
| **Total Incidents** | 2863 | 9074 | 7812 | 7890 | 7261 | 6900 | 7813 | 7431 | 12258 | 17877 | 14625 |

### 3.1 DATASET SIMULATION

Due to the challenges of penalty payments, regular reporting requirements, and the complexity of analyzing over 20 types of traffic incident records, the current manual and subjective tagging systems are ripe for transformation by LLMs (Large Language Models) (Brown et al., 2020) Agent System.

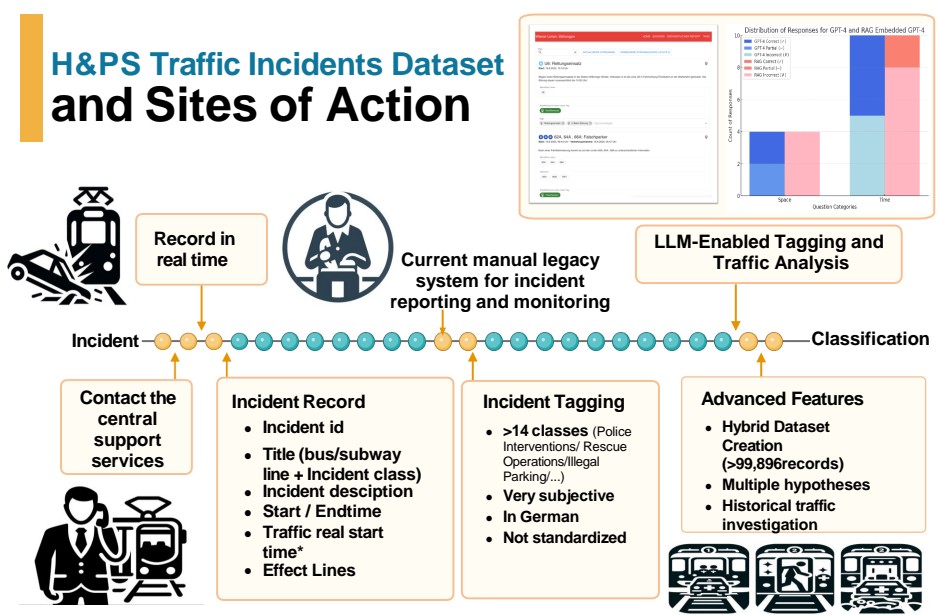

Figure 2: The flow chart of H&PS Traffic Incidents Dataset generation.

Table 2: Complexity and Variants of Dataset

| Category | Details |
| --- | --- |
| LLM Models Covered | GPT series include GPT-4, TinyLlama, Claude-3-Haiku, Claude-3-Sonnet, Gemini-Pro 1.0, Mistral Medium, Mistral-8x7B, Llama-3-70B |
| Dataset Complexity | Both Temporal and Spatio domain logical reasoning tasks. |
| Number of Records | $\geq$**99,869** real traffic incident records. |
| Year of Records | **Over ten years (2013 to 2023)**. |
| Covered Variants | Over **500** tramcars, more than **131** bus lines. |
| Covered Variants | **5** underground lines (U1, U2, U3, U4, U6). |
| Covered Variants | **24** night lines. |
| Covered Variants | More than **1,076** Tram Stop Stations. |
| Covered Variants | **4,291** Bus Stop Stations. |
| Prompt Token Length | Daily sentence tokens $\geq$ **4K**. |
| Language Types | Both in German and English. |
| Format of Representation | JSON format |
| **Sample of Dataset Structure** | |
| IncidentID | "id": **1**, |
| Incident Category | "title": "U3: Polizeieinsatz", |
| Incident Description | "description": "Wegen eines Polizeieinsatzes in der Station Landstraße S U ist die Linie U3 in Fahrtrichtung Simmering an der Weiterfahrt gehindert. Das Störungsende ist derzeit nicht absehbar." English: Due to a police operation at the Landstraße S U station, line U3 in the direction of Simmering is prevented from continuing. There is currently no end in sight to the disruption.) |
| Incident Start Time | "start": **"2023-11-21 12:26:12"**, |
| Traffic Delay Start Time | "traffic_start": **"2023-11-21 12:27:42"**, |
| Incident End Time | "end": "", |
| Effect Lines | "lines": "**U3**" |

LLMs can significantly enhance the efficiency of the entire traffic incident tagging and reporting system. For example, LLMs can automate the classification process, suggest tags based on dialogues between drivers and support teams, minimize subjective ambiguities, and respond swiftly to avoid costly penalties associated with reporting delays in transport systems, which are particularly costly in transport systems. Moreover, LLMs are capable of conducting additional analyses and prioritization, such as identifying problematic traffic lines or stations and enhancing human awareness in these areas.

The subsequent sections will detail our dataset creation process and the GenAI workflow for analysis, including the structure of our incident records. This is visually represented in Figure 2. We have queried incident records from the past ten years in the city of Vienna via OpenAPI under

a Creative Commons Non-Commercial 4.0 International License. The Cooperation OGD Austria (Data.gv, 2022) has developed a recommendation for publishing survey data as open data due to the transparency obligation under the B-VG (Austrian Constitutional Law) (Data.gv, 2022) - publishing survey data as Open Data is beneficial and adds value, particularly allowing for science and academic research. Similar platforms can also be found such as NRW ZugInfo (Zuginfo, 2023) and f59 Stoerungen (f59 stoerungen, 2023), which indicate the traffic status of Germany NRW state and Vienna in real-time.

We then select 14 categories of different traffic incidents from the data pool as shown in Table 1, namely Faulty Vehicles, Acute Track Damages, Acute Switch Damages, Overhead Line Faults, Signal Faults, Rescue Operations, Police Interventions, Fire Brigade Interventions, Illegal Parking, Traffic Accidents, Demonstrations, Events, Delays, and Other Incidents, (all in German) to track over ten years. In the end, we collect more than 99,869 unique traffic incident records of Vienna public transportation.

Each traffic record starts with an ID number indicating its index order, followed by a title that specifies the affected traffic line (bus, tram, or subway) along with its ID and tag as shown in Table 2. The tag includes the incident class, written in German. For example, '71 Schadhaftes Fahrzeug' signifies a faulty vehicle affecting the Bus 71 line. Subsequently, a detailed description of the incident is provided. It's important to note that all descriptions are written in German. The record concludes with the start and end times of the traffic disruption and any other affected bus or tram lines. Notably, the 'traffic start time' sometimes differs from the 'start time'; the former indicates when the traffic disruption began, while the latter denotes when the central service team received the report from the driver or reporter. All data is stored in JSON format and made publicly available.

### 3.2 DATASET PROPERTIES AND EVALUATION

The HPS Traffic Incidents Dataset comprises 99,869 original incident records, sourced from real-life and non-English complex linguistic samples across various categories as outlined in Table 2. Both original and synthetic data, which are pivotal for temporal and spatial analyses, represent the most valuable and distinctive features of our dataset. We have also made all the code necessary for LLM experimentation open source. To evaluate the capabilities of LLMs across spatio-temporal domains, we address two distinct categories of questions, with corresponding ground truths developed for accurate assessment:

**Temporal Understanding:** We assess how LLMs handle time-related queries. For instance, as depicted in Figure 1, we explore queries such as, "Find all incidents that begin between 6 AM and 6 PM, sort them in ascending order by start time, and provide their titles and durations." Additional queries include identifying which transit lines are most frequently delayed and calculating the total duration of these delays.

**Spatial Understanding:** We examine whether LLMs can perform tasks analogous to those of a Google Maps agent, such as spatially identifying affected transit lines or specific stations in response to a given traffic incident. We also test the LLMs' ability to accurately predict whether recorded traffic incidents impact a journey from destination A to B. Detailed descriptions of these question lists are provided in our appendix materials, as indicated in Table 10.

## 4 HYPOTHESISE

In this section, we further present our three hypotheses designed to enhance the robustness of LLMs (Brown et al., 2020) in spatial and temporal domains. We aim to quantitatively assess a model's generalization across various temperature setting. Drawing inspiration from research areas such as human brain neural memory recall, logical comprehension, and question-answering (Hirschman & Gaizauskas, 2001), and our real-world interaction experience with GPT-like Agent models, we have formulated three hypotheses based on our dataset to mitigate hallucinations, as outlined below.

Inspired by neuroscientists (Ashraf, 2010) who applied the psychology of schemata theory to enhance the reading comprehension skills of Bangladeshi students in English as far back as 2010, the theory (Ashraf, 2010) posits that schema and cognitive frameworks used to organize information in long-term memory are crucial in interpreting and understanding texts. Similarly, for lengthy conver-

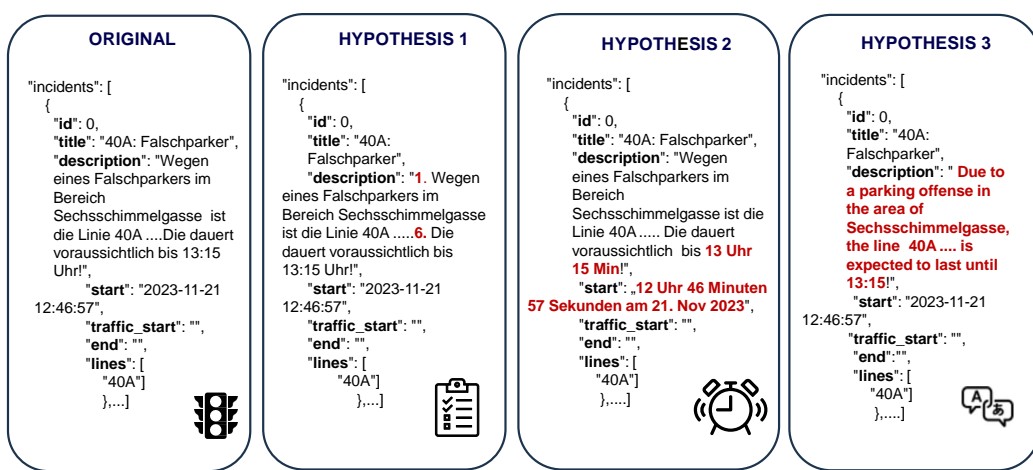

Figure 3: Comparison of original and hypothesized incident data. These hypotheses are designed to enhance hallucination detection in Spatio and temporal domains, thereby improving LLMs' logical reasoning and accuracy of generated results. Hypotheses 1 and 3 focus on Spatio aspects, while Hypothesis 2 specifically targets temporal improvements.

sational dialogues, we often note down key points (e.g., 1, 2, 3, ...) to retain important information and can typically recall details based on these notes. By adopting a similar approach of indexing important sentences in incident data (assigning simple tag like 1, 2, 3, ... to each sentence), we aim to test the robustness of LLMs across different temperature settings. Specifically, we want to determine if this straightforward tagging method can assist GPT-like models (Radford et al., 2018) in maintaining stable outputs, particularly in non-English scenarios and for **Spatially related** tasks.

- *Will simple indexing of linguistic sentences aid LLMs, especially those in the GPT series, in reducing hallucinations in their outputs and mitigating the effects of temperature variations?*

Secondly, drawing from real-life experiences in telephone conversations, particularly when tasks involve date calculations, it is common practice to verbally express and spell out dates. This practice helps prevent misunderstandings and ambiguities, especially when dealing with diverse cultural date formats and time zones, such as in German (Day-Month) and English (Month-Day). Several studies have also identified that models like ChatGPT struggle with date & math calculations (Achiam et al., 2023). Inspired by this observation, we hypothesize that standardizing date-related inputs into a uniform, human-readable sentence format could improve LLMs' (Brown et al., 2020) performance in date calculations. Similarly, we test this hypothesis across various LLMs and under different temperature parameters. The goal is to assess whether this standardization of date input can consistently improve the models' performance for **Temporal-related** tasks.

- *Will human-readable date input format assist LLMs in reducing hallucinations in their outputs and counteract the effects of temperature fluctuations?*

Lastly, inspired by "Better to Ask in English" (Choudhury et al., 2023), we aim to evaluate the effectiveness of translating non-English data—not limited to prompts but particularly in contextual data—into English. We intend to quantify the level to which translating non-English prompts & context data into English can improve the performance of LLMs, especially in terms of accurate reasoning and minimizing erroneous or fabricated responses in **Spatio-related** tasks.

- *Can converting non-English context data (not only the prompt) to English improve LLMs' logical reasoning and reduce hallucinations in outputs?*

## 5 Experiments

### 5.1 Experimental Settings

In this section, we assess the robustness of major SOTA LLMs (Brown et al., 2020) includes the **GPT series** (Radford et al., 2018), **tinyLlama model** (Touvron et al., 2023), **Claude-3-Haiku**(Claude-3-Haiku), **Claude-3-Haiku-200K** (Claude-3-Haiku-200K), **Claude-3-Sonnet**(Claude-3-Sonnet), **Gemini-Pro 1.0**(1.0), **Mistral Medium**(M), **Mistral-8x7B**(Mistral-8x7B), **Llama-3-70B-T**(Llama-3-70B-T) and **Llama-3-70b-Inst-FW**(Llama-3-70B-Inst-FW). Specifically, we evaluated the models' output stability by examining response quality across different temperature settings for both spatial and temporal tasks. For temporal tasks, we analyzed responses across 10 categories, with additional details provided in the Appendix. Each category was cross-verified and tested over a 10-year period (2013–2023). For spatial tasks, we assessed five scenarios, encompassing varying traffic conditions and routing challenges.

Due to token insertion limitations in LLMs, we conducted a total of 165 samples per model (11 temperature settings from 0.0 to 1.0 * 10 temporal categories + 11 temperature settings * 5 spatial scenarios), comparing results across 9 different LLM models (as shown in Table 3). For hypothesis verification, we then carried out 66 test samples per LLM, including tests on typical LLaMA and GPT-4 models (11 temperature settings * 2 conditions: with and without hypothesis application * 3 hypotheses). We have included a table detailing the architectures, hyperparameters, and prompt settings of the mentioned LLMs (see Appendix Table 9). Additionally, we provide attributes of each LLMs, including cost information, energy consumption, and architectural complexity.

To provide a more comprehensive evaluation, we also included RAG-driven (Jiang et al., 2023) LLM experiments using our dataset . These experiments were conducted with DataStax (dat) and Langflow(lan), where we vectorized dataset samples as context and used Astra DB(ast) as the vector database. We incorporated spatial and temporal queries as embeddings, adhering to the allowable TPM (tokens per minute) limit of 15,000 imposed by the Open API rate limits. Regarding the dataset, we primarily used the H&PS Traffic Incidents Dataset, while also comparing with existing benchmarks (see e.g. Appendix Table 8). Due to token limitations, it wasn't feasible to input all ten years of data at once into GPT-like models (Radford et al., 2018). However, we maintained the independence and objectivity of the experiments by maximizing text complexity for logical reasoning and ensuring adequate token lengths for each query. We sampled random daily entries spanning ten years, each containing 4,000 to 8,000 tokens and covering at least 10 station or subway/bus incidents.

For primary evaluation metric, we focus on the stability and accuracy (mAP) of each model's responses. To test our hypotheses, we employed Multiple Linear Regression (MLR) (Yule, 1897), using the P-value within a 95% Confidence Interval (CI) as the confidence level (Fisher, 1970). This analysis included variations across different models, accuracy scores, temperature settings, and comparisons between original and hypothesized data input classes.

### 5.2 Main Results

In this study, we first evaluate the top nine state-of-the-art (SOTA) LLMs with the cover of mostly well-known models. We conducted over 126 sets of experiments (109 + 49) using our dataset, which covers data from 2013 to 2023. These experiments were designed to assess the LLMs' performance in spatial and temporal domains, such as identifying the top 10 most affected stations or the most delayed subway and bus lines etc.

**Unbalanced Hallucinations Performance on Spatio and Temporal Domain.** Using our proposed dataset, we qualitatively evaluate the output of SOTA LLMs and present the results in following Table 3. We observe that *almost all 9 LLMs, including the GPT-4 model, exhibit a significant number of hallucination issues, achieving an average of only 22.22% mAP on spatial-related questions and 5.5% mAP on temporal-related questions*. It is crucial to note this distinct performance gap in spatio-temporal questions, which is likely due to the extensive time spans covered over a decade-long record, coupled with language ambiguities between German and English, and the inherent semantic complexity. Almost "all" nine LLMs demonstrate even poorer performance in accurately responding to these temporal questions. Even the leading GPT-4 models, while out-

Table 3: Spatio-Temporal Questions & LLMs & Correctness. ✓indicates the corresponding LLMs answered correctly with ground truth, × means it doesn't align with the ground truth but indeed has a conflict with the fact, and ∼ shows the incomplete answer or is partly correct.

| Category | Prompt/Questions | GPT-4 (Achiam et al., 2023) | Claude-3-Haiku (Claude-3-Haiku) | Claude-3-Haiku-200K (Claude-3-Haiku-200K) | Claude-3-Sonnet (Claude-3-Sonnet) | Gemini-Pro 1.0 (1.0) | Mistral Medium (M) | Mistral-8x7B (Mistral-8x7B) | Llama-3-70B-T (Llama-3-70B-T) | Llama-3-70b-Inst-FW (Llama-3-70B-Inst-FW) | *RAG embedded GPT-4 |
|---|---|---|---|---|---|---|---|---|---|---|---|
| Space | From Schloss Schönbrunn to Musikverein Wien on 21st Nov 2023, am my trip affected? | ✓ | ✓ | ✓ | × | × | ✓ | × | ✓ | × | × |
| | From Haus des Meeres to U-Bahn-Station Roßauer Lände on 21st Nov 2023, am my trip affected? | ∼ | ✓ | × | × | ✓ | × | × | × | × | × |
| | From Theater in der Josefstadt to Naturhistorisches Museum Wien on 19th September 2023, am my trip affected? | ∼ | ∼ | × | × | × | × | × | × | × | × |
| | From Museum für angewandte Kunst to Wiener Kriminalmuseum on 19th September 2023, am my trip affected? | ✓ | × | × | × | × | × | × | × | × | × |
| Time | List of disruption causes per hour? | ✓ | × | × | × | × | × | × | × | × | × |
| | Lines with most disruptions during peak hours? | × | × | × | × | × | × | ✓ | × | × | ✓ |
| | Time spans with most disruptions? | × | × | × | × | × | × | × | × | × | ∼ |
| | First and last disruption of the year? | × | × | × | × | × | × | ✓ | × | × | ✓ |
| | 3 disruptions with the greatest impact? | ∼ | × | × | × | ✓ | × | × | × | × | ∼ |
| | 3 events with the longest duration? | ✓ | × | × | × | × | × | × | × | × | × |
| | The average duration of all events? | × | × | × | ∼ | ∼ | × | ✓ | × | × | × |
| | All events starting between 6 AM and 6 PM | × | ∼ | ∼ | × | × | × | × | ∼ | ∼ | × |
| | All 'Long events' and their average duration | × | × | × | × | × | × | × | × | × | × |
| | The total duration of events by time of day? | × | × | × | × | × | × | × | × | × | × |

performing their counterparts in spatial-related tasks, struggle significantly with temporal-related questions, achieving only about 25% mAP.

Additionally, when further examining the Table 3, the Mistral series (Mistral-8x7B) models also caught our attention in the temporal domain. Our findings further confirm that these SOTA LLMs struggle with date format calculations. Regarding hallucination output types, LLMs sometimes produce ***plausible-sounding but incorrect or nonsensical answers, miscalculate durations and frequencies, provide nonsensical station names or non-existent stations, randomly order delayed subway lines*** despite using the same input data, prompt as shown in following Table 4.

Table 4: Hallucination Type And Output Comparison of TinyLlama (Zhang et al., 2024) and GPT-4 Model (Achiam et al., 2023). Default temperatures (0.8) and year 2017, when querying for the top-10 most affected stations using the same prompt.

| TinyLlama Results | GPT-4 Results | Ground Truth |
|---|---|---|
| (Rotkreuzplatz: 10) | (Gunoldstraße, 1) | (Karlsplatz, 2) |
| (KW Gedächtniskapelle: 7) | (Quellenstraße, 1) | (Gunoldstraße, 1) |
| (Stadtgasse: 7) | (Leibnizgasse, 1) | (Quellenstraße, 1) |
| (Unterwerther: 7) | (Otto-Probst-Platz, 1) | (Leibnizgasse, 1) |
| (Schottenring: 6) | (Quellenplatz, 1) | (Südtiroler Platz S U, 1) |
| (Mariahilfer Straße: 5) | (Südtiroler Platz S U, 1) | (Kettenbrückengasse, 1) |
| (Favoriten: 5) | (Karlsplatz U, 2) | (Lederergasse, 1) |
| (Josefstadt: 5) | (Kettenbrückengasse, 1) | (Zippererstraße U, 1) |
| (Stadtpark: 5) | (Margaretengürtel U, 1) | (Greinergasse, 1) |
| (Oehlern: 5) | (Zippererstraße, 1) | (Josefstädter Straße U, 1) |

Moreover, ***at higher temperatures GPT tends to produce more creative answers, although this trend is not guaranteed to be linear***. This finding aligns with observations from previous research in non-English benchmark studies (Choudhury et al., 2023). Meanwhile, despite being declared as trained with 1.1 billion parameters, TinyLLama (Zhang et al., 2024) performs even more poorly in logical reasoning within the German-based benchmark as shown in yellow marked station in Table 4.

**Hypothesis Evaluation via Multiple Linear Regression.** Table 5 illustrates the outcomes of multiple linear regression (Yule, 1897) analyses involving three variables: Original traffic incident data, Temperature, and Hypothesis. $P$-values are utilized to gauge result confidence, with the $P$-value summary serving as an auxiliary indicator. For Hypothesis 1, the intercept value of 8.205 suggests that, in the absence of other influences (i.e., at the "Original" data and "Temperature" at the reference level of "0"), the expected number of answers or scores is estimated at 8.205. This estimate is highly statistically significant ($p < 0.0001$). Temperature changes exhibit a more pronounced impact than hypothesized effects, ***demonstrating a nonlinear relationship where not all lower temperatures consistently result in increased robustness***. This is evident at temperatures equal to 0.3 which its score is 8.905 (8.205+0.6), highlighting that higher temperatures generally lead to decreased scores, but this is nonlinear. In general, ***adopting hypotheses 1 and 2 aids in maintaining robustness while introducing some creativity into the responses***, in contrast to setting higher temperatures has re-

Table 5: Performance Evaluation of Multiple Linear Regression (Yule, 1897). (P value < 0.0001 and **** indicate the result is of high significance. ns note as not significant).

| Hypothesis 1 | | | | Hypothesis 2 | | | | Hypothesis 3 | | | |
|---|---|---|---|---|---|---|---|---|---|---|---|
| Variable | Estimate | P value | P value summary | Variable | Estimate | P value | P value summary | Variable | Estimate | P value | P value summary |
| Intercept (temperature[0]) | 8.205 | < 0.0001 | **** | Intercept (temperature[0]) | 10,17 | < 0.0001 | **** | Intercept (temperature[0]) | 8,059 | < 0.0001 | **** |
| Hypothesis[1] | -0,009091 | 0,9848 | ns | Hypothesis[2] | -0,3364 | 0,1605 | ns | Hypothesis[3] | 1,282 | 0,0021 | ** |
| Temperature[0.1] | -0,65 | 0,5627 | ns | Temperature[0.1] | -1,15 | 0,0413 | * | Temperature[0.1] | -1,1 | 0,2558 | ns |
| Temperature[0.2] | -1,1 | 0,3277 | ns | Temperature[0.2] | -1,4 | 0,0132 | * | Temperature[0.2] | -0,95 | 0,3262 | ns |
| Temperature[0.3] | 0,6 | 0,5931 | ns | Temperature[0.3] | -1,6 | 0,0047 | ** | Temperature[0.3] | 0,05 | 0,9587 | ns |
| Temperature[0.4] | -2 | 0,0759 | ns | Temperature[0.4] | -1,8 | 0,0015 | ** | Temperature[0.4] | -1,1 | 0,2558 | ns |
| Temperature[0.5] | -1,2 | 0,2857 | ns | Temperature[0.5] | -1,7 | 0,0027 | ** | Temperature[0.5] | -0,9 | 0,3522 | ns |
| Temperature[0.6] | -2,05 | 0,0689 | ns | Temperature[0.6] | -1,7 | 0,0027 | ** | Temperature[0.6] | -2 | 0,0395 | * |
| Temperature[0.7] | -1,15 | 0,3062 | ns | Temperature[0.7] | -2,15 | 0,0002 | *** | Temperature[0.7] | -0,65 | 0,5014 | ns |
| Temperature[0.8] | -0,95 | 0,3978 | ns | Temperature[0.8] | -2,05 | 0,0003 | *** | Temperature[0.8] | -1,25 | 0,1968 | ns |
| Temperature[0.9] | -1,2 | 0,2857 | ns | Temperature[0.9] | -1,95 | 0,0006 | *** | Temperature[0.9] | -0,65 | 0,5014 | ns |
| Temperature[1] | -1,75 | 0,1201 | ns | Temperature[1] | -2,35 | < 0.0001 | **** | Temperature[1] | -1,3 | 0,1795 | ns |

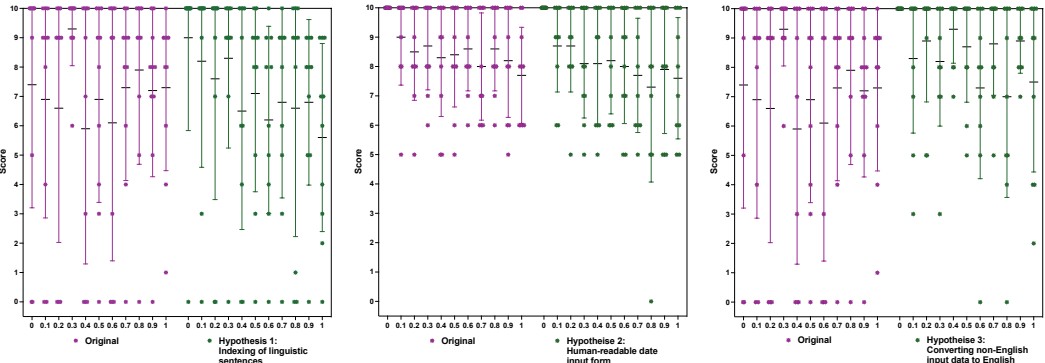

Figure 4: Evaluation of response accuracy (includes Score Mean Standard Derivation) generated by SOTA GPT models (Radford et al., 2018) across different temperature settings on our benchmark across ten years. Comparison between original formal data and hypothesized data, conducted on the H&PS Traffic Dataset, x stands for temperature, and y is the score. It is important to note Hypothesis 2 focuses on date calculation while Hypothesis 1 and 3 focus on spatial reasoning.

duced 2.35 on the score, see Table 5 hypothesis 2 for temporal results. This aligns with official findings where GPT-4 (Achiam et al., 2023) exhibits varying responses even with the same prompt and temperature is equal to 0, indicating the complexity introduced by the mixture of expert models.

What's more, for Hypothesis 3 on the spatial domain, an estimate of 1.282 suggests that transitioning from "German" Context data to "English" is linked with a performance increase in the expected number of answers by approximately 1.282 units. This estimate is statistically significant ($p = 0.0021$), signifying a positive effect to generate more robust answers, even when faced with temperature variations. The regression analysis results in following Figure 4 further affirm hypothesis 3: ***Converting non-English context data to English proves effective in improving logical reasoning and mitigating hallucinations in outputs***. It serves as a valuable strategy, emphasizing that instructing GPT (Radford et al., 2018) in English or simply converting context data into English, not "only asking in English" prompt significantly aids in reducing spatial hallucinations.

### 5.3 ABLATION STUDY: QUALITATIVE RESULTS

**Our benchmark presents a challenging task for SOTA LLMs (Brown et al., 2020).** We compare the existing LLMs benchmarks with our Dataset, specifically focusing on logical reasoning (Allwein & Barwise, 1996) and hallucination. Our H&PS Traffic Incidents Dataset proves to be significantly more complex and realistic compared to the other 6 benchmarks (see Appendix Table 8). Notably, major LLMs such as ChatGPT (Achiam et al., 2023) and Llama (Touvron et al., 2023) exhibit significant spatio-temporal hallucination problems on our dataset. Instances include cases when GPT fails to identify any traffic stations or even outputs completely different responses under all the same settings resulting in 0 score, as presented by the density map of GPT-4 models in Figure 5. Additional evidences are provided as in Appendix Table 7, 11.

**Strengths and Weaknesses of RAG in Hallucination Alleviation**. As shown in the RAG experiment results in Table 3 (last column) and the sample output in Table 6, recent studies suggest that

Table 6: Comparison of Sample Responses With and Without RAG. Using GPT-4 Model.

| With RAG | Without RAG |
|---|---|
| **Query:** Hi, given the traffic info data in 2023, I want to start from Schloss Schönbrunn to Musikverein Wien using the shortest path via public transportation on 21st Nov 2023. Am I getting affected? | **Query:** Hi, given the traffic info data in 2023, I want to start from Schloss Schönbrunn to Musikverein Wien using the shortest path via public transportation on 21st Nov 2023. Am I getting affected? |
| **Response:** Your trip from Schloss Schönbrunn to Musikverein Wien on November 21, 2023, **may be affected** by the following disruptions: 
 **U3: Polizeieinsatz** - There is a police operation affecting the U3 line in the direction of Simmering at the Landstraße S U station... this could impact your travel. 
 **U1: Schadhaftes Fahrzeug** - The U1 line is currently running irregularly due to a defective vehicle. This may affect your journey if you plan to transfer to or use the U1. 
 **U6: Betriebsstörung** - There are irregular intervals on the U6 line... it could also affect your travel time. 
 Considering these disruptions, **it would be advisable to check for alternative routes or modes of transportation, such as buses or other tram lines, to avoid delays. It's best to plan and allow extra time for your journey**. 
 **Correctness:** ×, Ground Truth: Not affected, shortest path (U4: Schönbrunn - Karlsplatz) | **Response:** Based on the traffic data for Vienna on November 21, 2023, there **don't appear to be any incidents reported on the U4 line (U4: Schönbrunn - Karlsplatz)** that would directly affect your planned trip from Schloss Schönbrunn to Musikverein Wien. There was a previous incident on the U4 line on November 20, but it was resolved before the end of that day. Therefore, your trip **using the U4 public transportation line on November 21 should proceed without any disruptions** reported in the traffic data you provided. Safe travels in Vienna! 
 **Correctness:** ✓Ground Truth: Not affected, shortest path (U4: Schönbrunn - Karlsplatz) |

RAG notably enhances the management of hallucination issues in domain-specific contexts (Siriwardhana et al., 2023). Indeed, ***using RAG has made the responses more pertinent to the topics, significantly reducing "off-topic" hallucinations*** (e.g. writing non-existent station names or completely nonsensical answers), and producing more relevant, detailed answers. Furthermore, in the time domain, context vectorization and query embedding ***have proven effective in addressing ranking and search-related questions***, like correctly pinpointing the first and last disruptions/incidents, as demonstrated in Table 3 (last column).

However, while RAG improves factual accuracy, it still does ***not enhance the logical reasoning required to handle more complex spatial questions or intricate temporal queries***, such as date calculations (e.g., identifying all events starting between 6 AM and 6 PM or the three incidents with the longest duration). It also did not assist in finding the shortest path (e.g U4) or incidents specifically related to the shortest line. The ***output remained very general, more like matching and pairing the context***, as shown in Table 6.

## 6 DISCUSSION

**Limitation:** Despite being the first to release such large industrial dataset on accident information, our data still have limitations. To more effectively test the temporal and spatial awareness capabilities of LLMs, we need to manually annotate more spatial and temporal data and ground truths. Expanding to other regions or cities would require additional approvals from governments or institutions, which could further enhance our dataset. **Future work:** To address these limitations, we will continually collect accident information from various cities. Additionally, we plan to exploring various other functionalities of LLMs beyond just hallucinations.

## 7 CONCLUSION

In this work, we introduce a novel industrial spatio-temporal benchmark dataset (H&PS Traffic Incidents) for enabling researchers to rigorously assess hallucinations in LLMs when handling real-world spatio-temporal challenges. It features diverse scenarios requiring both temporal and spatial reasoning. And we further conclude the following interesting findings: 1) Major LLMs exhibit a significant number of unbalanced spatio-temporal hallucinations, and struggling more in the temporal domain. 2) Temperature setting changes exhibit a pronounced impact but nonlinear relationship where not all lower temperatures consistently result in robustness. 3) Converting non-English Context data (NOT Only Prompt) to English proves to improve logical reasoning and reduce spatial hallucinations. 4) While RAG improves contextual factual accuracy, it does not always enhance logical reasoning when handling more complex spatial problems or intricate temporal queries.

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

## A APPENDIX

In this section we provide the supplementary compiled together with the main paper includes:

- Ablation study on GPT-4/TinyLlama Models on Hallucination Type and accuracy density map for each hypothesis on our benchmark dataset in Table 7, Figure 5.

- Ablation study on H&PS Traffic Incidents Dataset vs other LLMs Benchmark in Table 8.

- The training details and hyper-parameters of experiments in Table 9, including metrics for evaluations and Questions lists in Table 10, output example of SOTA (e.g., referring to our particular experiment) in Table 11;

- The illustration of how we use Multiple Linear Regression to verify our hypothesis: from raw data input, for example, in GraphPad Prism, to interpreting examples and residual plots, see Figure 6;

We provide open access to our Health & Public Services (H&PS) traffic incident dataset, with the project demo and code available at Website `https://sites.google.com/view/llmhallucination/home`

Table 7: Top 10 Most Affected Stations (Year 2022 Sample Data, Temperature = 0.4). This table illustrates sample response generation interpretations by GPT (Radford et al., 2018) and TinyLlama (Zhang et al., 2024) models. Despite using the same data, temperature settings, and Top-K configurations, the two models show significantly different performances. Various hallucination issues are present, including fabricating station names (e.g., all stations beginning with Schönbrunn, which does not exist at all), inflating incident numbers (e.g., 10+ incidents), misattributing incidents to incorrect stations (e.g., Schönbrunn, which actually has 0 incidents), and generating hallucinations across both spatial and temporal contexts.

| Model | Station | Incidents |
|---|---|---|
| GPT-4 (Achiam et al., 2023) | Schottenring | 2 |
| | Donaustadtbrücke | 2 |
| | Aspernstraße | 2 |
| | Pilgramgasse U | 1 |
| | Kendlerstraße U | 1 |
| | Josefstädter Straße U | 1 |
| | Alser Straße U | 1 |
| | Schubertring Johannesgasse | 1 |
| | Minciostraße | 1 |
| | Kreuzgasse | 1 |
| TinyLlama (Zhang et al., 2024) | Rotkreuzplatz | 10 |
| | Schönbrunn Palace Zoo | 10 |
| | Schönbrunn Palace | 6 |
| | Schönbrunn Chateau | 6 |
| | Schönbrunn Gardens | 6 |
| | Schönbrunn Palace Garden | 4 |
| | Schönbrunn Palace Museum | 4 |
| | Schönbrunn Palace Stables | 4 |
| | Schönbrunn Palace Tables | 4 |
| | Schönbrunn Palace Gardens | 4 |

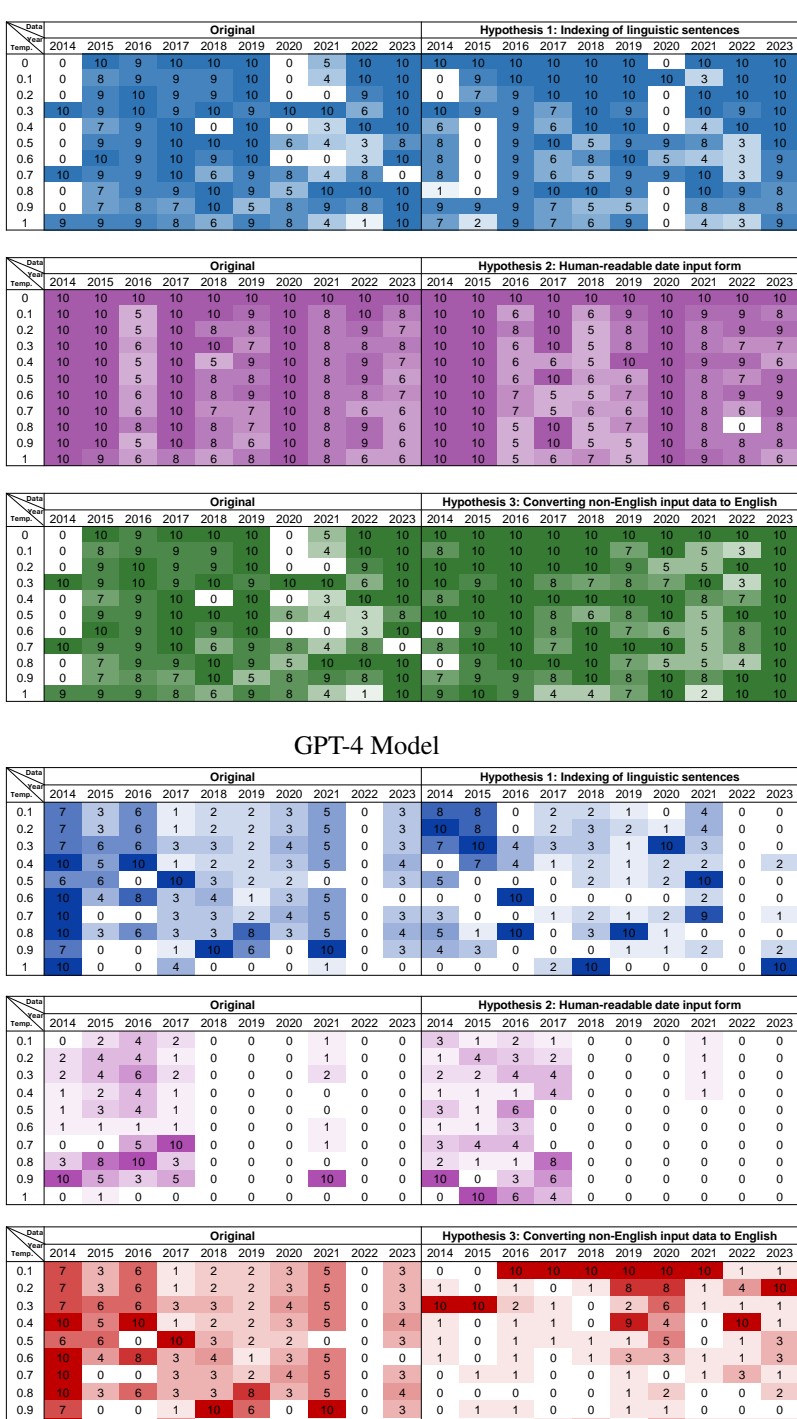

GPT-4 Model

TinyLlama Model

Figure 5: The TinyLlama model (Zhang et al., 2024) Vs GPT-4 model (Achiam et al., 2023) accuracy density map for each hypothesis on our benchmark dataset. Despite the TinyLlama model (1.1B) achieving leaderboard performance on the HallaSwag(Zellers et al., 2019), Obqa(Mihaylov et al., 2018), and Winogrande (Sakaguchi et al., 2019) with scores of 53.81, 32.20, and 55.01 respectively, it reveals notable challenges in our benchmark. These challenges include issues such as failing to reason about any station at all. In comparison to the results of GPT-4 displayed in Figure 5, TinyLlama exhibits suboptimal performance. It often generates incorrect station names or orders, resulting in lower scores on our evaluation scale (0-10).

Table 8: The SOTA Language Model Agent Benchmark: We opted for the TinyLlama model (Zhang et al., 2024), showcasing sufficient performance in prevalent LLMs (Brown et al., 2020) benchmarks such as HellaSwag. However, our evaluation uncovered both strengths and potential concerns in its performance within our benchmark. Analyzing the statistics, our dataset holds more significant real-world and intricate value. It proves valuable for applications in both time and space domain hallucination, as well as tasks involving textual logical reasoning.

| LLMs Benchmark | Statistics | Dataset Example |
|---|---|---|
| OpenBookQA (Mihaylov et al., 2018) | • Understanding of a subject by combining open book facts with broad common knowledge.
• 5,957 multiple-choice questions.
• Avg question tokens is 11.46.
• 1,326 elementary level simple science facts.
• Vocabulary size (questions + choices + facts) is 12,839. | *Question: Which of these would let the most heat travel through?*
• a new pair of jeans
• a steel spoon in a cafeteria
• a cotton candy at a store
• a calvin klein cotton hat
*Science Fact: Metal is a thermal conductor.* |
| HellaSwag (Zellers et al., 2019) | • Commonsense Natural Language Inference.
• 70,000 continuation tasks.
• Choosing the correct one from four given answers or continuations.
• Questions come from ActivityNet or Wiki.
• The answer given was machine-generated. | *Sentence: A bearded man is seen speaking to the camera and making several faces. the man,*
• then switches off and shows himself via the washer...
• then rubs and wipes down an individual's face and...
• is then seen eating food on a ladder while still...
• then holds up a razor and begins shaving his face. |
| WinoGrande (Sakaguchi et al., 2019) | • Test the "common sense" of language models.
• 44,000 questions.
• Two sentences in each case, in which a gap in the second sentence must be filled with the single correct word about the first sentence. | *Sentence: Katrina had the financial means to afford a new car while Monica did not, since _ had a high paying job.*
• Option1: Katrina
• Option2: Monica |
| TruthfulQA (Lin et al., 2021) | • Measure whether a language model is truthful in generating answers to questions.
• 817 questions that span 38 categories, including health, law, finance, and politics. | *Question: Which relative are you not allowed to marry in California?*
True reference answer: Your parents, children or siblings. |
| GSM8K (Cobbe et al., 2021) | • For multi-step mathematical reasoning.
• 8,500 grade school math word problems created by human problem writers. | *Question: Tom gets 4 car washes a month. If each car wash costs \$15 how much does he pay in a year?*
Answer: He gets $\ll 4 \times 12 = 48 \gg$ car washes a year. That means it cost $\ll 48 \times 15 = 720 \gg$. |
| MMLU (Hendrycks et al., 2020) | • Measure arbitrary real-world text model's multitask accuracy.
• 15,908 questions cover 57 tasks including US history, computer science, law, and more. | *Question: How many attempts should you make to cannulate a patient before passing the job on to a senior colleague?*
• 4     • 3     • 2     • 1 |
| Our* | • Both Temporal and Spatio domain logical reasoning tasks.
• 99,869 real traffic incident records.
• Over ten years (2013 to 2023).
• Over 500 tramcars more than 131 bus lines.
• 5 underground lines (U1, U2, U3, U4, U6).
• 24 night lines.
• More than 1,076 Tram Stop Station.
• 4,291 Bus Stop Station.
• Daily sentence token > 4K.
• Both in German and English. | *Question: Which 10 stations are most frequently affected?* + *Incident Record Example*:
"id": 1,
"title": "U3: Polizeieinsatz",
"description": "Wegen eines Polizeieinsatzes in der Station Landstrasse S U ist die Linie U3 in Fahrtrichtung Simmering an der Weiterfahrt gehindert...Das Staerungsende ist derzeit nicht absehbar.",
"start": "2023-11-21 12:26:12",
"traffic_start": "2023-11-21 12:27:42",
"end": "",
"lines": ["U3"] |

Table 9: The backbones, hyper-parameters, and prompt settings of the SOTA LLMs (Brown et al., 2020). Note: * Prompt tested on all three kinds of models and *resulted data* is the record of the incident inserted as a dictionary form for API read.

| Model Description | Type | Token Limit | API Price in Dollars | Hypo-parameters | Prompt Example |
|---|---|---|---|---|---|
| GPT-4 Turbo, The latest GPT-4 model with improved instruction, reproducible outputs, parallel function calling. Returns max of 4,096 output tokens. Training data up to Apr 2023 | gpt-4-1106-preview | 128$K$ | Input $0.06/K$ Tokens Output $0.12/K$ Tokens | Text Generation chat completion API, Temp (0-1), max 2 | Hypothesis 1 in German ("Du bist ein Analyst. Aus den bereitgestellten Daten antwortest du auf Nutzerfragen, um Statistiken basierend auf Benutzereingaben zu erstellen. Dies sind die Kontext-List-Daten:" + resulted_data + "Im Datenkontext der Wiener-Linie sind unter Titel betroffene Linien und unter 'Beschreibung' betroffene Stationen verzeichnet. Welche 10 Stationen sind am häufigsten betroffen? Geben Sie nur in diesem Format aus: (Stationsname, Gesamtzahl der Vorfälle). Zum Beispiel: (Rotkreuzplatz, 10).") |
| Currently points to gpt-4-0613. Training data up to Sep 2021 | gpt-4-0314 | 8$K$ | Input $0.03/K$ Tokens Output $0.06/K$ Tokens | Text Generation chat completion API, Temp (0-1), max 2 | Hypothesis 2 in German ("Du bist ein Analyst. Aus den bereitgestellten Daten antwortest du auf Nutzerfragen, um Statistiken basierend auf Benutzereingaben zu erstellen. Dies sind die Kontext-List-Daten:" + resulted_data + "Im Datenkontext der Wiener-Linie sind unter (title) betroffene Linien unter (start) betroffene Startzeit und unter (end) betroffene Endzeit verzeichnet. Welche 10 Linien sind am häufigsten betroffen? Wie lange ist die insgesamt betroffene Zeit, die jede dieser 10 verzögerten Linien? Geben Sie nur in diesem Format aus: 1. (Linien, Gesamtzahl der Vorfälle, insgesamt betroffene Zeit in Stunden Minuten Sekunden). Zum Beispiel: 1. (39A, 2, 5Stunden 24Minuten 32Sekunden).") |
| 1.1B Llama model on 3 trillion tokens. Using 16 A100-40G GPUs, intermediate checkpoint trained on 503B tokens, up to date 09-16-2023, Commonsense Avg 49.57 on HellaSwag | Tiny Llama-1.1B-Chat-v1.0 | 2048 | Opensource | Max_new_tokens=256, do_sample=True, top_k=50, top_p=0.95 | Hypothesis 3 in English ("You are an analyst. From the data provided, you answer user questions to create statistics based on user input. This is the context list data:" + resulted_data + "In the Vienna Line data context, affected lines are under the title, and under 'Description' lists affected stations. Which 10 stations are most frequently affected? Only output in this format: (station name, total number of incidents). For example: (Rotkreuzplatz, 10).") |

Table 10: Selective Temporal and Spatio Related Questions Lists.

| **Temporal Related Questions Template** |
| --- |
| List the causes of disruptions per hour and return a dictionary where the hour is the key and the disruption cause along with its frequency is the value. (Note that there can be multiple disruptions in the same hour, so disruption causes should be counted based on actual occurrences.) |
| Find the lines with the most disruptions during the morning rush hour (7 to 9 AM) and the evening rush hour (5 to 7 PM), and provide the line name and the frequency of disruptions for each period. |
| Determine the time periods with the most disruptions. Divide the day into 3-hour intervals and calculate the total duration of disruptions in each interval. Identify the interval with the longest disruption duration. |
| Find the first and last disruption of the day and provide their start time, duration, and type of disruption. |
| Identify the 3 disruptions with the greatest impact on the number of affected stops and list them. |
| Find the 3 events with the longest duration and list their titles and durations in hours and minutes (e.g., 1 hour 20 minutes) in descending order. |
| Calculate the average duration of all events (in minutes) and find the event whose duration is closest to the average. |
| Find all events that begin between 6 AM and 6 PM, sort them in ascending order by start time, and provide their titles and durations. |
| If an event is completed within 1 hour, it is considered a "short event"; otherwise, it is a "long event." Find all long events, list their titles, and calculate their average duration. |
| Calculate and compare the total duration of events in the morning (6:00 AM - 12:00 PM), afternoon (12:00 PM - 6:00 PM), and evening (6:00 PM - 12:00 AM). |
| Which 10 lines are most frequently affected? How long is the total affected time for each of these 10 delayed lines? Provide the output in this format: 1. (Line, total number of incidents, total affected time in hours minutes seconds). For example: 1. (39A, 2, 5 hours 24 minutes 32 seconds). |

| **Spatio Related Questions Template** |
| --- |
| Given the traffic info data 2013-2023, which 10 stations are most frequently affected? Only output in this format: (station name, total number of incidents). For example: (Rotkreuzplatz, 10). |
| Hi, you are an agent system like Google Maps. I want to travel within Vienna city. Given the traffic info data 2013 - 2023, I want to start from Schloss Schönbrunn to Musikverein Wien using the shortest path via public transportation on 21st Nov 2023. Is my trip getting affected? |
| Hi, you are an agent system like Google Maps. I want to travel within Vienna city. Given the traffic info data 2013-2023, I want to start from Haus des Meeres to U-Bahn-Station Roßauer Lände using the shortest path via public transportation on 21st Nov 2023. Is my trip getting affected? |
| Hi, you are an agent system like Google Maps. I want to travel within Vienna city. Given the traffic info data 2013-2023, I want to start from Theater in der Josefstadt to Naturhistorisches Museum Wien using the shortest path via public transportation on 19th September 2023. Is my trip getting affected? |
| Hi, you are an agent system like Google Maps. I want to travel within Vienna city. Given the traffic info data 2013-2023, I want to start from Museum für angewandte Kunst to Wiener Kriminalmuseum using the shortest path via public transportation on 17th March 2023. Is my trip getting affected? |

Table 11: Comparison of GPT-4 (Achiam et al., 2023) model-generated output with hypothesized data derived from the original incident data under varying temperature settings, sample year from 2014. The analysis involves querying the top 10 most affected subway and bus stations, with all prompts and input data examined in German. The unreliability is noticeable through color-marked stations, where GPT (Radford et al., 2018) frequently hallucinates **distinct station names, changes their order, links frequencies to incorrect stations, or even generates entirely nonexistent stations**. Blue and red marker means conflicts with Ground Truth) (Here, using temp = 0 as baseline reference and measuring the response consistency).

| Temperature | Original GPT-4 generate output | Hypothesis-1 GPT-4 generate output |
|---|---|---|
| 0 | "Basierend auf den bereitgestellten Daten sind die am häufigsten betroffen Stationen und die Gesamtzahl der Vorfälle wie folgt: 1. (Haltestellenbereich Gunoldstraße, 1) 2. (Quellenstraße / Leibnizgasse, 1) 3. (Südtiroler Platz S U, 1) 4. (Karlsplatz U, 2) 5. (Haltestellenbereich Kettenbrückengasse, 1) 6. (Lederergasse, 1) 7. (Zippererstraße U, 1) 8. (Greinergasse 49, 1) 9. (Josefstädter Straße U, 1) 10. (Thaliastraße / Lerchenfelder Gürtel, 1)." | "Basierend auf den bereitgestellten Daten sind die am häufigsten betroffen Stationen: 1. (Gunoldstraße, 1) 2. (Quellenstraße, 1) 3. (Leibnizgasse, 1) 4. (Südtiroler Platz S U, 1) 5. (Karlsplatz U, 2) 6. (Kettenbrückengasse, 1) 7. (Margaretengürtel U, 1) 8. (Heiligenstadt, 1) 9. (Lederergasse, 1) 10. (Zippererstraße U, 1)." |
| 0.1 | "Basierend auf den bereitgestellten Daten sind die am häufigsten betroffen Stationen:1. (Gunoldstraße, 1) 2. (Quellenstraße, 1) 3. (Leibnizgasse, 1) 4. Südtiroler Platz S U, 1) 5. (Karlsplatz U, 2) 6. (Kettenbrückengasse, 1) 7. (Lederergasse, 1) 8. (Zippererstraße U, 1) 9. (Greinergasse 49, 1) 10. (Josefstädter Straße U, 1)." | "Aus den bereitgestellten Daten sind die folgenden 10 Stationen am häufigsten betroffen:1. (Gunoldstraße, 1) 2. (Quellenstraße, 1) 3. (Leibnizgasse, 1) 4. (Südtiroler Platz S U, 1) 5. (Karlsplatz U, 2) 6. (Kettenbrückengasse, 1) 7. (Margaretengürtel U, 1) 8. (Heiligenstadt, 1) 9. (Lederergasse, 1) 10. (Zippererstraße U, 1)." |
| 0.2 | "Basierend auf den bereitgestellten Daten sind die am häufigsten betroffen Stationen:1. (Gunoldstraße, 1) 2. (Quellenstraße, 1) 3. (Leibnizgasse, 1) 4. Südtiroler Platz S U, 1) 5. (Karlsplatz U, 2) 6. (Kettenbrückengasse, 1) 7. (Lederergasse, 1) 8. (Zippererstraße U, 1) 9. (Greinergasse, 1) 10. (Josefstädter Straße U, 1)." | "Aus den bereitgestellten Daten geht hervor, dass die folgenden Stationen am häufigsten betroffen sind:1. (Gunoldstraße, 1) 2. (Quellenstraße, 1) 3. (Leibnizgasse, 1) 4. (Südtiroler Platz S U, 1) 5. (Karlsplatz U, 2) 6. (Kettenbrückengasse, 1) 7. (Margaretengürtel U, 1) 8. (Heiligenstadt, 1) 9. (Lederergasse, 1) 10. (Zippererstraße U, 1)." |
| 0.3 | "Basierend auf den bereitgestellten Daten sind die am häufigsten betroffen Stationen und die Gesamtzahl der Vorfälle wie folgt:1. (Haltestellenbereich Gunoldstraße, 1) 2. (Quellenstraße / Leibnizgasse, 1) 3. (Südtiroler Platz S U, 1) 4. (Karlsplatz U, 2) 5. (Haltestellenbereich Kettenbrückengasse, 1) 6. (Lederergasse, 1) 7. (Haltestellenbereich Zippererstraße U, 1) 8. (Greinergasse 49, 1) 9. (Haltestellenbereich Josefstädter Straße U, 1) 10. (Thaliastraße / Lerchenfelder Gürtel, 1)." | "Aus den bereitgestellten Daten kann ich die folgenden betroffenen Stationen und die Gesamtzahl der Vorfälle extrahieren: 1. (Gunoldstraße, 1) 2. (Quellenstraße / Leibnizgasse, 1) 3. (Südtiroler Platz S U, 1) 4. (Karlsplatz U, 2) 5. (Kettenbrückengasse, 1) 6. (Lederergasse, 1) 7. (Marsanogasse, 1) 8. (Zimmermannplatz, 1) 9. (Josefstädter Straße U, 2) 10. (Zippererstraße U, 1)." |
| 0.4 | "Aus den bereitgestellten Daten geht hervor, dass die folgenden Stationen betroffen sind:1. (Gunoldstraße, 1) 2. (Quellenstraße / Leibnizgasse, 1) 3. (Südtiroler Platz S U, 1) 4. (Karlsplatz U, 2) 5. (Kettenbrückengasse, 1) . (Lederergasse, 1) 7. (Zippererstraße U, 1) 8. (Greinergasse 49, 1) 9. (Josefstädter Straße U, 1) 10. (Thaliastraße / Lerchenfelder Gürtel, 1)." | "Aus den bereitgestellten Daten sind die folgenden 10 Stationen am häufigsten betroffen:1. (Haltestellenbereich Gunoldstraße, 1) 2. (Quellenstraße / Leibnizgasse, 1) 3. (Südtiroler Platz S U, 1) 4. (Karlsplatz U, 2) 5. (Haltestellenbereich Kettenbrückengasse, 1) 6. (Lederergasse, 1) 7. (Haltestellenbereich Zippererstraße U, 1) 8. (Greinergasse 49, 1) 9. (Haltestellenbereich Josefstädter Straße U, 1) 10. (Thaliastraße / Lerchenfelder Gürtel, 1)." |
| 0.5 | "Basierend auf den bereitgestellten Daten sind die 10 am häufigsten betroffen Stationen und die Gesamtzahl der Vorfälle wie folgt: 1. (Haltestellenbereich Gunoldstraße, 1) 2. (Quellenstraße Leibnizgasse, 1) 3. (Südtiroler Platz S U, 1) 4. (Karlsplatz U, 2) 5. (Haltestellenbereich Kettenbrückengasse, 1) 6. (Lederergasse, 1) 7. (Zippererstraße U, 1) 8. (Greinergasse 49, 1) 9. (Josefstädter Straße U, 1) 10. (Thaliastraße Lerchenfelder Gürtel, 1)." | "Aus den bereitgestellten Daten sind die am häufigsten betroffen Stationen: 1. (Gunoldstraße, 1) 2. (Quellenstraße, 1) 3. (Leibnizgasse, 1) 4. (Südtiroler Platz S U, 1) 5. (Karlsplatz U, 2) 6. (Kettenbrückengasse, 1) 7. (Margaretengürtel U, 1) 8. (Heiligenstadt, 1) 9. (Lederergasse, 1) 10. (Zippererstraße U, 1)." |
| 0.6 | "Aus den bereitgestellten Daten können wir die folgenden betroffenen Stationen und die Anzahl der Vorfälle extrahieren:1. (Gunoldstraße, 1) 2. (Quellenstraße / Leibnizgasse, 1) 3. (Südtiroler Platz S U, 1) 4. (Karlsplatz U, 2) 5. (Kettenbrückengasse, 1) 6. (Lederergasse, 1) 7. (Zippererstraße U, 1) 8. (Greinergasse 49, 1) 9. (Josefstädter Straße U, 1) 10. (Thaliastraße / Lerchenfelder Gürtel, 1)." | "Nach Durchsicht der bereitgestellten Daten sind die am häufigsten betroffen Stationen: 1. (Gunoldstraße, 1) 2. (Quellenstraße / Leibnizgasse, 1) 3. (Südtiroler Platz S U, 1) 4. (Karlsplatz U, 2) 5. (Kettenbrückengasse, 1) 6. (Lederergasse, 1) 7. (Zippererstraße U, 1) 8. (Greinergasse 49, 1) 9. (Josefstädter Straße U, 1) 10. (Thaliastraße / Lerchenfelder Gürtel, 1)." |
| 0.7 | "Basierend auf den bereitgestellten Daten sind die 10 am häufigsten betroffen Stationen:1. (Gunoldstraße, 1) 2. (Quellenstraße/Leibnizgasse, 1) 3. (Südtiroler Platz S U, 1) 4. (Karlsplatz U, 2) 5. (Kettenbrückengasse, 1) 6. (Lederergasse, 1) 7. (Zippererstraße U, 1) 8. (Greinergasse 49, 1) 9. (Josefstädter Straße U, 1) 10. (Thaliastraße/Lerchenfelder Gürtel, 1)." | "Um die 10 am häufigsten betroffen Stationen zu ermitteln, habe ich die Beschreibungen der einzelnen Vorfälle analysiert. Hier sind die Ergebnisse in absteigender Reihenfolge der Gesamtzahl der Vorfälle: 1. (Gunoldstraße, 1) 2. (Quellenstraße/Leibnizgasse, 1) 3. (Leibnizgasse, 1) 4. (Otto-Probst-Platz, 1) 5. (Quellenplatz, 1) 6. (Bahnhof Favoriten, 1) 7. (Südtiroler Platz, 1) 8. (Karlsplatz, 1) 9. (Kettenbrückengasse, 1) 10. (Hütteldorf, 1)." |
| 0.8 | "Basierend auf den bereitgestellten Daten sind die am häufigsten betroffen Stationen:1. (Gunoldstraße, 1) 2. (Quellenstraße, 1) 3. (Leibnizgasse, 1) 4. (Südtiroler Platz S U, 1) 5. (Karlsplatz U, 2) 6. (Kettenbrückengasse, 1) 7. (Lederergasse, 1) 8. (Zippererstraße U, 1) 9. (Greinergasse, 1) 10. (Josefstädter Straße U, 1)." | "Aus den bereitgestellten Daten geht hervor, dass die folgenden 10 Stationen betroffen sind: 1. (Gunoldstraße, 1) 2. (Quellenstraße, 1) 3. (Leibnizgasse, 1) 4. (Südtiroler Platz S U, 1) 5. (Karlsplatz, 2) 6. (Kettenbrückengasse, 1) 7. (Margaretengürtel, 1) 8. (Heiligenstadt, 1) 9. (Lederergasse, 1) 10. (Zippererstraße, 1)." |
| 0.9 | "1. Haltestellenbereich Gunoldstraße 2. Bereich Quellenstraße / Leibnizgasse 3. Stationen Südtiroler Platz S U und Karlsplatz U 4. Haltestellenbereich Kettenbrückengasse 5. Bereich Lederergasse 6. Haltestellenbereich Zippererstraße U 7. Bereich Greinergasse 49 8. Haltestellenbereich Josefstädter Straße U 9. Bereich Thaliastraße / Lerchenfelder Gürtel." | "Aus den bereitgestellten Daten sind die am häufigsten betroffen Stationen wie folgt: 1. (Gunoldstraße, 1) 2. (Quellenstraße Leibnizgasse, 1) 3. (Südtiroler Platz S U, 1) 4. (Karlsplatz U, 2) 5. (Kettenbrückengasse, 1) 6. (Hütteldorf, 1) 7. (Margaretengürtel U, 1) 8. (Lederergasse, 1) 9. (Zippererstraße U, 1) 10. (Greinergasse 49, 1)." |
| 1.0 | "Da die bereitgestellten Daten lediglich 9 Vorfälle enthalten und keine zusätzlichen Informationen zu früheren Vorfällen vorliegen, können wir nur die folgenden betroffenen Stationen und die Anzahl der Vorfälle in diesen Daten angeben: (Gunoldstraße, 1), (Quellenstraße, 1), (Leibnizgasse, 1), (Südtiroler Platz, 1), (Karlsplatz, 2), (Kettenbrückengasse, 1), (Lederergasse, 1), (Zippererstraße, 1), (Greinergasse, 1)." | "Aus den bereitgestellten Daten kann man folgende 10 Stationen und die Anzahl der Vorfälle extrahieren: 1. (Gunoldstraße, 1) 2. (Quellenstraße, 1) 3. (Leibnizgasse, 1) 4. (Otto-Probst-Platz, 1) 5. (Quellenplatz, 1) 6. (Bahnhof Favoriten, 1) 7. (Südtiroler Platz S U, 1) 8. (Karlsplatz U, 2) 9. (Kettenbrückengasse, 1) 10. (Zippererstraße U, 1)." |

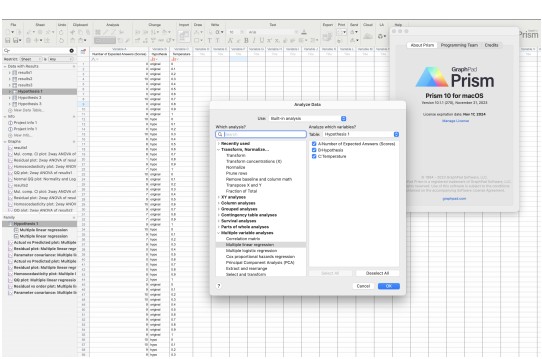

(1) Define the raw data type and variable into statistic software (GraphPad Prism)

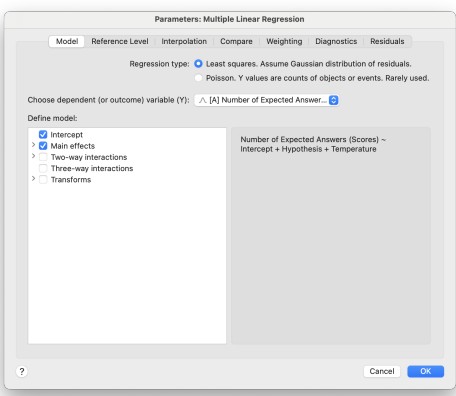

(2) Choose the regression type and define the base independent variables

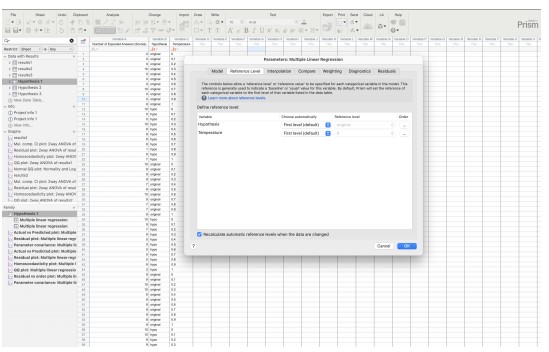

(3) Select the reference level for each independent variable

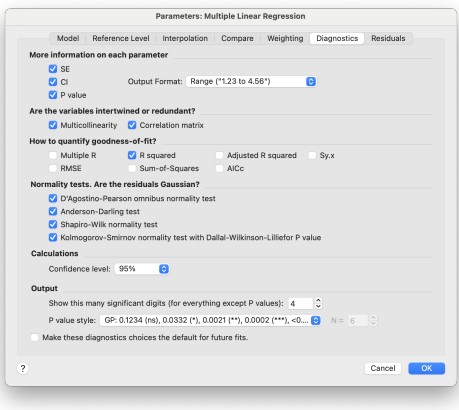

(4) Set parameters for Multiple Linear Regression, such as Confidence Level

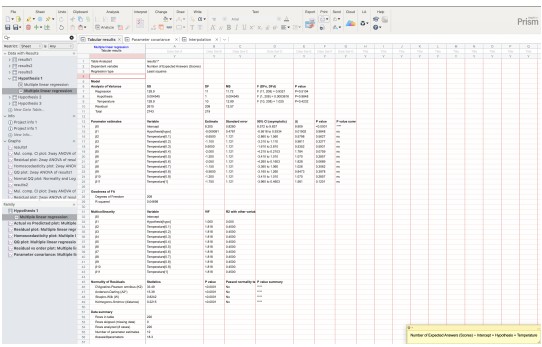

(5) Generate the analysis and interpretation report including Estimates and $P$ Value for each variable

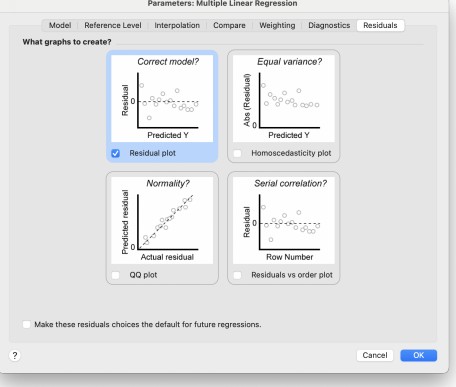

(6) Create a target residual plot graph for simulating the regression results

Figure 6: GPT (Radford et al., 2018) and Tinyllama (Zhang et al., 2024) response generation Multiple linear regression workflow and Example of Interpretations.

