# OpenReview forum: "Industrial Benchmarking of LLMs: Assessing Hallucination in Traffic Incident Scenarios with a Novel Spatio-Temporal Dataset"
_ICLR.cc/2025/Conference — ICLR 2025 Conference Withdrawn Submission_

### Official Review · Reviewer_mX3h · 2024-10-31

**Soundness:** 2
**Presentation:** 2
**Contribution:** 2
**Rating:** 3
**Confidence:** 3

**Summary:**

This paper introduces a new Health & Public Services (H&PS) dataset called H&PS Traffic Incident Dataset, which contains approximately 100,000 traffic incident reports written in German over the past 10 years. The authors then tested the capabilities of the current state-of-the-art (SOTA) large language models (LLMs) on several tasks based on this dataset, with a focus on spatio-temporal question answering. Most of the LLMs tested performed poorly on the dataset, although the authors verified that certain techniques were effective in improving performance. These techniques included (1) numbering sentences sequentially in the description, (2) spelling out dates and times, (3) translating the description into English, and (4) using Retrieval Augmented Generation (RAG).

**Strengths:**

* The paper publishes and open-sources a dataset in a non-English language. The dataset covers real-world, socially important data, namely traffic incident reports.

**Weaknesses:**

* Unclear tasks and presentation: although the paper presents a new dataset and benchmark, the very definition of what tasks the paper is trying to solve with LLMs remain unclear. The abstract says "domain of traffic incident classification" and Section 3.1 mentions "tagging systems" but Section 5.2 mentions "identifying the top 10 most affected stations" and throughout the paper the task seems to be generic question-answering based on the examples.
  * Related, but the paper is unclear how the test data used in the experiments is created
  * If the task is question-answering (for example, list of affected stations, as shown in Table 3) how do you evaluate the results? The paper mentions "we focus on the stability and accuracy (mAP)" but that's not how you usually evaluate question answering (also mAP is precision, not accuracy, and used for evaluating information retrieval in general). Figure 4 is also drawn using "score" which is not defined in the paper.
  * More importantly, many of the questions, such as "Find all incidents that begin between ... " (in Section 3.2) are simple information retrieval tasks that can be solved by simple SQL. Given that LLMs lack full access to the entire database (except in limited cases with RAG), the task appears ill-suited for LLMs.
* Trivial improvements: the paper proposes some research questions and techniques to improve the task proposed in the paper (Section 4). All of these seem trivial and some have shown to work in other domains already (e.g., translating into English), so they don't constitute the core technical contribution of the paper.
  * Slightly related, but I'm not sure why the experiments are overly focused on the temperature parameter. Can we simply choose the best parameter based on, e.g., the validation set and use it throughout?
* Societal impact: Finally, even though the dataset is derived from the data that is socially important, there's no strong backing why the publish dataset and the tasks addressed in the paper is important socially.

**Questions:**

* Section 1. "across multiple languages" — I believe the dataset is monolingual (just German)?
* Section 2. "ChatGPT (Achiam et al., 2023) — the first author of the ChatGPT paper is "OpenAI"
* Section 2. "minority languages like German" — I wouldn't personally call German "a minority language" if that's the case almost all languages in the world, except English and maybe Mandarin, would be minority languages...
* Section 3.2. I'm not sure what the authors mean by "a Google Maps agent." Does Google Map have an agent? What does it do?
* Section 4 "Hypothesise" you meant "Hypotheses"?
* It'd be helpful to include some statistics about the evaluation data used for the experiments.
* Can you improve LLMs for this particular task by continued pretraining (or fine-tuning) on this dataset?

---

### Official Review · Reviewer_CpPg · 2024-11-04

**Soundness:** 2
**Presentation:** 2
**Contribution:** 3
**Rating:** 3
**Confidence:** 3

**Summary:**

The paper studies the spatiotemporal reasoning capability of LLMs. It argues that existing LLM benchmarks still lack data and evaluation on spatial and temporal reasoning; there are not enough findings regarding this reasoning capability. The paper introduces a dataset and benchmark for handling spatiotemporal traffic incident questions. The dataset includes 99,869 traffic incident records from Vienna’s public transport system across 14 incident categories for the past 10 years. In addition, it also includes both German and English languages.

They test three hypotheses related to sentence numbering, standardized date-to-text conversion, and translating German context data to English on 15 (or 14) questions on varying LLMs and temperatures. Additionally, they examine the application of Retrieval-Augmented Generation (RAG) to mitigate hallucinations. In summary, the results suggest that none of the LLMs can effectively answer spatial or temporal questions, even with RAG. In addition, the multiple-regression analysis shows that the hypothesis of translating to English significantly improves the performance and that temperatures have a non-linear relationship with the performance.

**Strengths:**

1. The paper provided a dataset that consisted of spatial and temporal information.
2. The paper confirmed the previous findings that LLMs have unsatisfactory performance in spatiotemporal reasoning when using standard prompt engineering techniques (including RAG).

**Weaknesses:**

1. While introducing a new dataset is a welcome contribution, the introduced data quality is not thoroughly validated. In Section 3, the paper discusses dataset creation (or simulation) with a minimum discussion on the "LLM-Enabled Tagging and Traffic Analysis" process in Figure 2. In addition, most existing benchmarks have questions and ground-truth answers (see a few examples in Table 8 in the appendix), and the introduced dataset only consists of records. The evaluation of LLMs using the proposed benchmark is then unclear.
2. The paper conflates the definition of hallucination and equates it to low mAP (line 373). There could be multiple reasons why the answer is incorrect; not all will be plausible.
3. I found that many statements in the text lack support from the results. For example,
    1. Table 3 shows the experiment results for 14 questions, but the text (line 339) suggests 15 (10 + 5).
    2. Table 3 shows RAG's rather poor performance, but the text (line 505) suggests that it reduces "off-topic" hallucinations.
    3. Table 4 shows "ns" for Hypothesis 1 and Hypothesis 2, but the text (line 431) suggests that "... [the hypotheses] aids in maintaining robustness".
4. There are many papers that work in enhancing spatio-temporal reasoning capability of LLMs that the paper should acknowledge and discuss. For examples,
    1. Li, F., Hogg, D. C., & Cohn, A. G. (2024, March). Advancing spatial reasoning in large language models: An in-depth evaluation and enhancement using the stepgame benchmark. In Proceedings of the AAAI Conference on Artificial Intelligence (Vol. 38, No. 17, pp. 18500-18507).
    2. Ouyang, X., & Srivastava, M. (2024). LLMSense: Harnessing LLMs for High-level Reasoning Over Spatiotemporal Sensor Traces. arXiv preprint arXiv:2403.19857.
    3. Wu, W., Mao, S., Zhang, Y., Xia, Y., Dong, L., Cui, L., & Wei, F. (2024). Mind’s eye of LLMs: Visualization-of-thought elicits spatial reasoning in large language models. arXiv preprint arXiv:2404.03622.
    4.

**Questions:**

-

**Details Of Ethics Concerns:**

The release of public traffic incidents might include personal information. The authors should discuss how privacy is being considered and preserved.

---

### Official Review · Reviewer_jHvA · 2024-11-11

**Soundness:** 1
**Presentation:** 2
**Contribution:** 2
**Rating:** 3
**Confidence:** 4

**Summary:**

This paper introduces a traffic incidents spatio-temporal dataset about spatio-temporal QA instances in German. The data is processed based on publicly available dataset. Evaluation of multiple LLM families is performs on this proposed dataset. It also shows the ablation results of several prompting techniques, like whether includes bullet point index, human-readable data input format and English-only query.

**Strengths:**

* The proposed task and dataset is highly relevant to the real-world problem. The setting is different from synthetic datasets. It is very realistic and would be helpful if the dataset setting is sound.
* A rich ablation study and analysis of the evaluation result

**Weaknesses:**

* The evaluating test set is limited. A small number of test cases are used, it's hard to say whether the produced evaluation result can be generalized.
* The setting is not well explained, leaving concerns about the soundness of the proposed setup
	* What is the exact input provided to the model (besides the question prompt)? The questions are heavily depend on whether related incidents are provided and how they are provided. We don't have sufficient details in the current version of paper.
	* How is the metric mAP calculated
	* For the RAG baseline, what is the size and pool used for retrieval?
* Using generation output under different temperatures to represent model generalizability and robustness is a questionable setting
* Though the paper mentioned that the existing benchmarks do not support multilingualism well, this proposed dataset is also not broad in terms of language coverage since all descriptions are in German. It's not bad to only have one language, but a smaller-scale claim would be helpful.

**Questions:**

For the interpretation of the RAG performance (the paragraph before Section 6), why is the conclusion that it enhances the reasoning while it answers more questions incorrectly compared with plain GPT-4 in Table 3?

Please also refer to the question mentioned in the weaknesses section.

---

### Note · Authors · 2024-11-16

**Comment:**

## **Summary**
We sincerely thank the reviewers for their valuable feedback and appreciate the time spent reviewing our paper. We are grateful for the recognition of the strengths and value of our dataset. Our dataset comprises nearly 99,869 authentic traffic incident records from Vienna, provided in collaboration with OGD Austria as part of the open data initiative, aligned with the transparency obligations under the Austrian Constitutional Law (B-VG). This dataset is particularly significant for scientific and academic research.

We are pleased to note that the reviewers acknowledged the **novelty and relevance** of our dataset. Reviewer **jHvA** highlighted that our "real-world dataset, distinct from synthetic datasets", offers valuable insights from actual accident scenarios. Additionally, the "comprehensive case studies and ablation analyses" presented rich evaluations across numerous LLMs, demonstrating the utility of our dataset (Reviewer **jHvA**). Reviewer **CpPg** acknowledged the provision of "spatial and temporal data" and our detailed comparison using RAGs on LLMs. Reviewer **mx3h** noted that our dataset "covers socially relevant data for LLM" research and mentioned its "open-source availability" for the ML community.


After thorough deliberation with our co-author team and a deep review of the feedback, we have decided to **withdraw our submission** for this round of ICLR. We sincerely thank the reviewers for their time and insights. While we may not fully agree with all the points raised, we respect the feedback and are committed to further improving our manuscript. Therefore we made further effort to clarify the reviewer’s feedback.

## **Evaluation of Test Dataset**: (Reviewer jHvA)
We have addressed this aspect clearly in our paper, specifically in Section 5.1 (lines 328–345). We conducted two types of experiments: temporal and spatial. For temporal queries, we sampled over 10 classes, as detailed in Appendix Table 10, with each class tested across a 10-year span (2013–2023). Inputs and prompts were adjusted annually. For spatial queries, we tested five different scenarios, including various high/low traffic areas and different routing questions across multiple years. Given token insertion constraints in LLMs, we executed over 160 samples per LLM, comparing nine models. For hypothesis verification (testing if certain hypotheses reduce hallucinations), we varied temperature settings ( 0 to 1) and used three types of questions, with unique samples tested across the 10-year span. The detailed logs of these comprehensive tests are provided in the Appendix and on our GitHub, [Website](https://sites.google.com/view/llmhallucination/home).  We believe this is a sufficient number of test samples given the complexity of the dataset and the computational cost involved.

## **Input Setting and RAG Input**: (Reviewer jHvA)
For both temporal and spatial experiments, Tables 9 and 10 outline the types of questions injected into the LLMs, along with the relevant traffic records for context (**indicated in Table 9, column 6**). For evaluating RAG in an industry context, the **entire 2023 year traffic record** was vectorized using Astra DB as the vector database. We incorporated spatial and temporal queries as embeddings, adhering to the TPM (tokens per minute) limit of 15,000 (Open API rate limits). The details are accessible through our GitHub logs and website as well.

## **Language Coverage, Ground Truth, and mAP Interpretation**: (Reviewer mX3h)
Our dataset includes both German and English data. We **did not** claim that "current benchmarks are inadequate"; rather, we highlighted the complexity and multilingual coverage of our dataset, as "most existing SOTA datasets are synthetic or focus on scientific Q&A without comprehensive spatio-temporal attributes". Ground truth data for temporal and spatial queries were manually calculated and are available in the attached Excel file on our website and GitHub (see **Table 4, column 3** and **Table 3 caption**). The **scores** were calculated based on the match with the ground truth.

## **Advanced Findings on RAG**: (Reviewer mX3h)
While previous works [1-4] acknowledge certain limitations of RAG (e.g., difficulty contextualizing retrieved data or handling non-English queries), our experiments provided new insights. We found that RAG enhanced relevance but did not improve logical reasoning in spatial-temporal domains. Although RAG helped with query-context pairing, it failed to identify specific incidents related to the shortest U4 line, indicating **general matching rather than precise reasoning**. Our findings on RAG’s performance in temporal and spatial domains are novel contributions *not previously documented* in RAG studies.

| **Category** | **Prompt/Questions** | **GPT-4 Response Accuracy** | **RAG-embedded GPT-4 Accuracy** |
|--------------|----------------------|-----------------------------|----------------------------------|
| **Space**    | From Schloss Schönbrunn to Musikverein Wien on 21st Nov 2023, is my trip affected? | ✓ | ✗ |
|              | From Haus des Meeres to U-Bahn-Station Roßauer Lände on 21st Nov 2023, is my trip affected? | ~ | ✗ |
|              | From Theater in der Josefstadt to Naturhistorisches Museum Wien on 19th September 2023, is my trip affected? | ~ | ✗ |
|              | From Museum für angewandte Kunst to Wiener Kriminalmuseum on 19th September 2023, is my trip affected? | ✓ | ✗ |
| **Time**     | List of disruption causes per hour? | ✓ | ✗ |
|              | Lines with most disruptions during peak hours? | ✗ | ✓ |
|              | Time spans with most disruptions? | ✗ | ~ |
|              | First and last disruption of the year? | ✗ | ✓ |
|              | 3 disruptions with the greatest impact? | ✗ | ~ |
|              | 3 events with the longest duration? | ✓ | ✗ |
|              | The average duration of all events? | ✓ | ✗ |
|              | All events starting between 6 AM and 6 PM | ✗ | ✗ |
|              | All 'Long events' and their average duration | ✗ | ✗ |
|              | The total duration of events by time of day? | ✗ | ✗ |

### Legend
- **✓**: Ground Truth (GT)
- **✗**: Output completely missed the point
- **~**: Half-Truth (partially correct response)

The results were generated using the GPT-4-0 model, with vectorized samples from the 2023 dataset context.

## Selective RAG Response

**Question:**
> Hi, given the traffic info data in 2023, I want to start from Schloss Schönbrunn to Musikverein Wien using the shortest path via public transportation on 21st Nov 2023. Am I getting affected?

**Response:**
> Your trip from Schloss Schönbrunn to Musikverein Wien on November 21, 2023, may be affected by the following disruptions:
> - **U3**: _Polizeieinsatz_ - There is a police operation affecting the U3 line in the direction of Simmering at the Landstraße S U station, with no clear end to the disruption. Since you may need to use the U3 to reach Musikverein Wien, this could impact your travel.
> - **U1**: _Schadhaftes Fahrzeug_ - The U1 line is currently running irregularly due to a defective vehicle. This may affect your journey if you plan to transfer to or use the U1.
> - **U6**: _Betriebsstörung_ - There are irregular intervals on the U6 line due to operational disruptions. If your route involves this line, it could also affect your travel time.

> Considering these disruptions, it would be advisable to check for alternative routes or modes of transportation, such as buses or other tram lines, to avoid delays. It's best to plan and allow extra time for your journey.

**Correctness:** ✗


## **Clarifications on Reviewers' Misunderstanding**: (Reviewer CpPg)
We thank *Reviewer CpPg* for pointing out specific areas for improvement. We will emphasize key statements to avoid misunderstandings in future revisions. For instance, the *LLM-enabled tagging and traffic analysis* in Figure 2 were only included as a part of our GenAI application for industry use in traffic incident classification (as shown on our website). Additionally, the statement on lower mAP scores does not imply hallucination; rather, lower mAP scores are used to examine LLM results across nine models. Generally, if an LLM does not hallucinate, it is expected to achieve better matches with the ground truth.

The temporal scenarios (10) and spatial scenarios (5) covered lines U1, U3, U4, U6, and one bus case scenario. We acknowledge that there may be omissions in Table 3, where only four specific questions are shown in the space column. We will enhance this table's clarity in future work.

Regarding Hypotheses 1 and 2, we aimed to maintain robustness, as indicated in Table 5. While lower temperature settings are generally believed In this community to yield more stable results, our findings show that using hypotheses improved robustness beyond simply adjusting temperature settings. and this “maintaining robustness is compared to set temperature, **it has context**. (As line 428 and line 431. seen revision)

## **Additional References and Citations**: (Reviewer CpPg)
We appreciate the suggestions from Reviewers mx3h and CpPg. While it was not feasible to cover all recent benchmarks in our comparisons, we included six benchmarks and the latest temporal or spatial works, such as UrbanGPT 2024. We will incorporate additional these recent work in our future updates.

We respect the review process, and we look forward to revising our manuscript and resubmitting when a more thorough and fair evaluation can be ensured.

Best regards,

The authors

[1] Jiang, Zhengbao, et al. "Active retrieval augmented generation." arXiv preprint arXiv:2305.06983 (2023).
[2] Karpukhin, Vladimir, et al. "Dense passage retrieval for open-domain question answering." arXiv preprint arXiv:2004.04906 (2020).
[3] Guu, Kelvin, et al. "Retrieval augmented language model pre-training." International conference on machine learning. PMLR, 2020.
[4] Siriwardhana, Shamane, et al. "Improving the domain adaptation of retrieval augmented generation (RAG) models for open domain question answering." Transactions of the Association for Computational Linguistics 11 (2023): 1-17.

**Withdrawal Confirmation:**

I have read and agree with the venue's withdrawal policy on behalf of myself and my co-authors.